# The Speaker Method: A Novel Release Method for Offspring Mammals and 5-Year Study on Three Costa Rican Mammals

**DOI:** 10.3390/ani13233669

**Published:** 2023-11-27

**Authors:** Encarnación García-Vila, Roger Such, Bárbara Martín-Maldonado, Elena Tarròs, Elisa L. Sorribes, Cristina Calvo-Fernandez

**Affiliations:** 1Jaguar Rescue Center, Punta Cocles, Limón 70403, Costa Rica; encargarciavila@gmail.com (E.G.-V.); roger_such@hotmail.com (R.S.); etarrosjrc@gmail.com (E.T.); 2Department of Veterinary Medicine, School of Biomedical and Health Sciences, Universidad Europea de Madrid, 28670 Villaviciosa de Odón, Spain; bmmjimenezvet@gmail.com; 3Veterinary Faculty, Universidad Complutense de Madrid, 28040 Madrid, Spain; elisa.lopez.sorribes@gmail.com; 4Research Group for Food Microbiology and Hygiene, DTU National Food Institute, Kemitorvet, 204, 2800 Kongens Lyngby, Denmark; 5Research Group for Foodborne Pathogens and Epidemiology, DTU National Food Institute, Kemitorvet, 204, 2800 Kongens Lyngby, Denmark

**Keywords:** *Alouatta palliata*, *Bradypus variegatus*, *Choloepus hoffmanni*, rehabilitation, release, rescue center, wildlife, conservation

## Abstract

**Simple Summary:**

This study presents the Speaker Method as a novel approach for wildlife offspring release to facilitate their reunion with their mothers in their natural habitats, thereby avoiding the need for captive rearing. This method uses call records of the offspring to attract their mothers effectively. In this context, we aimed to prove the Speaker Method’s efficacy in releasing the offspring of three mammal species that arrived at a wildlife rescue center. The study showed promising results, successfully releasing 45.8% of Hoffmann’s two-toed sloths, 91.9% of brown-throated sloths, and 50% of mantled howler monkeys. These results provide empirical effectiveness for the Speaker Method as a release technique for offspring, underscoring its superiority over conventional nursery care by humans facing inherent challenges in rearing young animals separated from their maternal sources.

**Abstract:**

Nowadays, wild animals are threatened by humans, with the number of species and individuals decreasing during recent years. Wildlife rescue centers play a vital role in the conservation of wildlife populations. This study aims to describe a new release technique, the Speaker Method, to rescue and facilitate the reunion of different baby mammals that arrived at a wildlife rescue center with their mothers within their natural habitat, avoiding the need for captivity. This method is based on a recorded baby’s cry played on a speaker to make a “call effect” in the mother. The efficacy of the Speaker Method for babies’ reunion with their mothers was 45.8% in Hoffmann’s two-toed sloths (*Choloepus hoffmanni*) and 91.9% in brown-throated sloths (*Bradypus variegatus*). Among the mantled howler monkeys (*Alouatta palliata*), 50% of the babies could be released using this new technique. The findings suggest that the method could be helpful in the early release of young individuals, highlighting higher release outcomes in these three species compared to traditional nursery care provided by human caretakers, who face inherent difficulties in raising young animals without their mothers.

## 1. Introduction

Global biodiversity is decreasing precipitously, leading to the extinction of many species. According to the Living Planet Index (LPI), 69% of monitored wildlife populations have decreased from 1970 to 2018, and overall, Latin America shows the most significant regional decrease, with a 94% average population abundance decline [1]. Costa Rica harbors almost 4% of global biodiversity [2] and forms part of the Mesoamerican Biological Corridor, which is considered a hotspot [3]. Despite its richness in animal and vegetal biodiversity, 2.2% of animal species are critically endangered, 5.9% endangered, and 9.2% have vulnerable status [4]. Most of them are threatened by human activities (urbanization, deforestation, increase in agricultural lands, habitat fragmentation, hunting games, etc.), like the Hoffmann’s two-toed sloth (*Choloepus hoffmanni*), the brown-throated sloth (*Bradypus variegatus*), and the mantled howler monkey (*Alouatta palliata*) [4]. Moreover, some of these are unique indigenous species with critical ecological niches, such as the sloths; thus, the care and protection of their wild populations are essential for conservation.

Sloths are considered keystone species for ecosystem conservation since their diet, mainly folivores, plays a crucial role in the nutrient cycle of forests and, therefore, in conserving vegetal biodiversity. Moreover, their mutualism with moths and algae is a unique ecological interaction [5]. The number of sloths arriving at wildlife rescue centers (WRCs) has steadily increased recently, with approximately 400 sloths being admitted annually in both major WRCs in the South Caribbean region [6]. In fact, according to the IUCN Red List of Threatened Species, Hoffmann’s two-toed sloth and the brown-throated sloth are listed as least concern and decreasing [4].

The mantled howler monkey, for its part, is threatened mainly by habitat fragmentation. Its dietary flexibility has helped it to survive habitat fragmentation [7]. However, changes in feeding behavior entail adverse effects on individuals, such as changes in the microbiota or insufficient energy for regular activity, and on the ecosystem, since the more fragmented the habitat, the greater the concentration of monkeys in the tree canopies, which leads to further degradation of the habitat [7,8]. The last update of the IUCN Red List of Threatened Species classified the mantled howler monkey as vulnerable and decreasing [4].

In this context, WRCs play an essential role in animal welfare, treating and caring for injured, sick, and stray native animals to release them back into their habitats when rehabilitated [9]. Nevertheless, after rescue and rehabilitation efforts, the release success rate is approximately 50% for animals admitted to WRCs from various global regions [10,11]. A recent study published in 2023 showed that although the release rate in recovery centers exceeds 50%, for mammals it is slightly lower (44.8%) [12]. Another recent publication about Costa Rican WRCs stated that the release rate of one of them, at 36%, below the world average, might be due to material/human resources or to the characteristics of the native species [13]. A large proportion of admissions into these centers are “orphaned” animals that have been found without their parents. Even though those orphans do not present lesions and have a good health status at their arrival at the WRC, their care is complicated, and their prognosis is not always favorable. They require specific attention with special care facilities, alongside more time and resources than a healthy older animal [9,14]. In addition, they are particularly susceptible to diseases and other problems related to living with humans, such as behavioral issues like taming, which may result in the animal being unable to return to the wild [15,16]. Then, although it is possible to breed and reintroduce orphaned animals into the wild successfully, it is preferable, whenever possible, to return them to their mother. Moreover, their offspring stage with their group is essential for correct learning and behavioral development. Individuals deprived of these interactive contacts may show motor, biological, and behavioral deficiencies [17]. This is why developing an early-release method for these healthy young animals is essential. 

Different techniques have been applied to reintroduce animals into the wild depending on the species, age, and particular situation of the animal, e.g., whether it has been raised in a WRC or not, soft releases, hard releases, and short-term soft releases (or semi-hard releases). The latter is most suitable for captive-bred individuals [18]. Techniques to increase the survivability of captive-bred individuals after release are being studied. Reintroduction has been achieved even after rejection of the offspring, as in the case of the wooly monkey (*Lagothrix lagotricha*), which could be returned to its mother after the dominant male separated them [19]. In raptors, Miller [20] developed a method of successfully reuniting hatchlings with their mothers using recorded calls that has been successfully employed in several species. The process is based on raptors’ strong parenting bonds and the trend to stay near the nest for a time after losing their nestlings. The recorded calls of nestlings should stimulate their parents to respond, confirming their presence in the area and making reunification possible [20]. This method could be helpful to mammals. Still, to our knowledge, there are yet to be any publications about a validated technique of recorded calls to mothers for reunification with offspring.

In this context, this study aimed to describe an early-release technique based on recorded calls of offspring, the Speaker Method. Also, we assessed the success of the method in the release of offspring from a WRC of three different mammal species of Costa Rica (the Hoffmann’s two-toed sloth, the brown-throated sloth, and the mantled howler monkey) during a five-year period. 

## 2. Materials and Methods

### 2.1. Ethical Approval

The research methodology outlined in this article adhered rigorously to the approved guidelines stipulated by the Jaguar Rescue Center’s (JRC) (Punta Cocles, Limón 70403, Costa Rica) management plan in alignment with the resolutions outlined in ACLAC-DRFVS-PVS-002-2018 following the MINAE regulations.

### 2.2. Study Area and Animal Species

The study data were collected from animals admitted to the JRC Wildlife Hospital, located in Punta Cocles, near Puerto Viejo de Talamanca, Limón Province, Costa Rica (9°38′22.0″ N, 82°43′14.8″ W), which receives hundreds of animals yearly [21]. All the cases included in the present study were reported in the same protected areas: the Gandoca–Manzanillo Wildlife Refuge and the Cahuita National Park (Figure 1).

Hoffmann’s two-toed sloth is found mainly in lowland and montane tropical forests, both deciduous and mixed-deciduous [22]. These sloths are relatively solitary, the sex ratio is female-biased, and male territory ranges vary from 1.1 to 139.5 ha [22]. The range size and natural dispersal habits differ significantly from one individual to another, and their herbivore–omnivore diet consists mainly of leaves, fruits, insects, and small vertebrates [23]. 

The brown-throated sloth is the most widespread of the three-toed sloths [24]. In contrast to the two-fingered sloths, they are solitary, their diet consists only of leaves, and they do not cover vast territories [25]. 

The mantled howler monkey can inhabit several distinct vegetation environments, including mature evergreen forests, deciduous and riparian forests, mangroves, and anthropogenically disturbed forests [26,27]. Although they spend most of their time resting (65–74%), the rest is spent collecting food in the tree canopies [28]. Group sizes vary considerably (up to more than 40 individuals), but the average group size is 15.2 [29].

### 2.3. Release Strategy: The Speaker Method

#### 2.3.1. The Rescue

When local people call the center’s emergency number after observing an animal in danger, a trained rescue team from the JRC goes to the place from where the call was made, takes care of the animal, and brings it to the custody of the JRC following an established protocol according to the species. Before returning rescued animals to our center, our team takes special care when dealing with young animals to ensure their well-being. If the rescued animal is a juvenile, our protocol involves a thorough check to determine if the mother is still in the rescue area. The confirmation of the mother’s identity relies solely on visual monitoring. Sloths provide a higher level of security due to their limited daily movement and solitary behavior. Considering that this method allows the release of offspring on the same day of collection, the distance covered by the mother during that time is short. However, in the case of monkeys, the group dynamics and their continuous and fast movement make it more challenging to identify the biological mother, even though it is known that they follow fixed routes. On the other hand, in captivity, it has been observed that howler monkey females can adopt orphaned offspring that enter the center, suggesting the ability of this species to adapt under free-living conditions. 

Once the animals have been safely transported to the center, a comprehensive health assessment is conducted by a veterinarian to ascertain their health condition for immediate release into their natural habitat. The veterinary examination also facilitates the detection of any anomalies associated with maternal rejection. Notably, such incidents predominantly involve accidental falls, typically concerning viable offspring. An integral component of this process is inserting microchips for individual identification, a critical step before proceeding with the immediate release. Otherwise, the individual remains at the center of its recovery for a quarantine period (for this work, quarantine was considered to be the first 40 days during which all rescued animals were in the hospital or nursery facilities with intensive care and veterinary treatment). The survival probability of these offspring species in the wild is minimal, so their early return to the mother may be vital in rehabilitating and reintroducing these animals into ecosystems. If the mother was visible at the rescue, the vet and rescue teams discussed whether they could return the baby to the mother or not. Alternatively, if the animal presented conditions consistent with survivability, the Speaker Method with the mother was carried out.

#### 2.3.2. The Speaker Method

The Speaker Method is a technique of reunion of a rescued baby with its mother. It starts with annotating and photographing the spot where the rescued animal has been found. After the medical check-up, if the health status is optimal, the baby must cry to record the sound, so the method conducted is species-dependent. In the case of sloths, gentle movements make the babies cry for seconds; otherwise, the monkeys usually start crying when they do not have contact with their mothers.

The release time depended on the species, with daylight times being optimal for the brown-throated sloth and the mantled howler monkey, while for Hoffmann’s two-toed sloth, the nighttime was better. Once at the rescue location, the speaker was deposited at the base of the tree, but if it was a noisy place near a road or the sea, the speaker could be placed up a tree in a basket, as close to the mother as possible. Regardless of the mother’s presence, the recorded cry was played on a loudspeaker to produce a “call effect.” The speaker’s volume had to be regulated depending on the distance from the mother, and it was essential not to let her hear the baby crying simultaneously so as not to confuse her. To prevent this, as the mother came closer to the meeting point, the volume decreased, and when she was at a distance from which she could hear her baby’s actual calls, the speaker was turned off. If the mother was observed to show interest in the claim (i.e., presented nervous behavior, began to descend from the trees, emitted vocalizations, etc.), the team called the JRC to reunite the baby with the mother. The speaker should be replaced by the real baby when visual contact occurs quickly. Silence is fundamental throughout the process for its success. As the mother may have moved, the team wandered the area with the speaker, using binoculars to look for animals on the branches that could be the mother. Using blankets, baskets, or other objects to avoid the mother’s attack is dangerous and not recommended, nor is workers climbing the tree to approach the offspring, as the mother may feel threatened and attack both the offspring and the staff. A ladder can be used instead to bring the branch with the baby closer to the mother. It may take a few minutes to recognize the baby. If it is during the night, spotlights may be necessary. In such a case, they should never focus on the mother directly, and they should be used only if strictly required (e.g., when there is poor visibility, such as in dense forest areas away from towns).

The reunion process might not be possible if conducted by inexperienced people, leading to potentially dangerous situations, for example, when presenting the baby to a male. Some mistakes that could produce unsuccessful attempts are related to excess sound, artificial light such as a flashlight, movement speed, or people around the zone. For howler monkeys and three-toed sloths, sexual identification can be achieved by observing size differences. Additionally, male three-toed sloths can be distinguished by a brightly colored patch of fur on their upper back [30]. However, in Hoffmann’s two-toed sloths, sexual differentiation is challenging from a distance, as the external genitalia are only discernible at close proximity [30]. Males prioritize attacking whatever they have in front of them, even the baby. Because of their poor sight, they are guided by smell, and therefore, they would likely smell the JRC team before the baby, and the encounter could transform into aggression. Other dangerous situations could occur if the potential mother were a juvenile female that then drops the baby, if the mother becomes stressed and attacks the baby by mistake, or if a different female with other babies or a pregnant animal is attracted by the cry, which has been observed during the years of work in the JRC in sloths and monkeys. This underscores the importance of vigilance for the absence of aggressive or hostile behaviors (Figure 2) (Appendix A).

### 2.4. Data Collection and Processing

All the data about the rescue and procedures performed in the JRC from the last five years (January 2018 to December 2022) were collected with an informatic database specially designed for the center, including incomes, outcomes, days in the center, animal parameters (weight, age, etc.), clinical information, veterinary procedures, and release success. The “age of admission” was considered relevant in this study, dividing the animals into three categories defined by veterinarian criteria: baby, juvenile, and adult. Only those animals considered “babies” have been included in this study. 

The Speaker Method was considered effective when the mother came to the reunion point in response to the recorded call played with the speaker and accepted the baby. The release team stayed at this point for a few minutes or even hours to ensure the acceptance. If the baby was rejected, the application of the method was considered unsuccessful, and the baby returned to the JCR. As the babies were released with an individual microchip (T-VAS glass microchip), the JCR would be notified if they were later rejected (and the baby was admitted to another WRC) or found dead.

If release attempts using the Speaker Method are unsuccessful, the offspring remain in rehabilitation and undergo a developmental process until they reach a stage of viability as adults for release. This prolonged captivity period might, regrettably, entail risks to their survival. Survivability and mortality rates during rehabilitation were also analyzed and compared. 

## 3. Results

### 3.1. Descriptive Statistics of the JRC

From January 2018 to December 2022, 964 sloths were reported to have come into the JRC: 535 Hoffmann’s two-toed sloths and 429 brown-throated sloths. Overall, only 315 were considered babies: 192 Hoffmann’s two-toed sloths and 123 brown-throated sloths.

The weight of the Hoffmann’s two-toed sloth babies ranged from 194 g to 1300 g, with a mean of 520.88 ± 32.89 g. In the case of the brown-throated sloth, the weight of babies ranged from 176 g to 1100 g, with an average weight of 492.81 ± 43.08 g. Considering the data found in the database, we can state that the average weight of the offspring was higher in Hoffmann’s two-toed sloths than in brown-throated sloths. 

For mantled howler monkeys, 254 individuals were rescued by the JRC during these five years, where 95 were considered “babies.” The average weight was 584.12 ± 66.48 g, with a maximum weight of 1626 g and a minimum of 223 g.

### 3.2. Captive Survivability

Regardless of age, a release was considered successful when the animal was set free, monitored through visual observation, and did not re-enter the JRC or any other WRC. The rates were 45.9% of brown-throated sloths (197/429), 40% of Hoffmann’s two-toed sloths (214/535), and 37% of mantled howler monkeys (95/254) (Table 1). Survivability in captivity, and thus successful release, was much higher in adults than in babies for sloths; the rates were 28% for brown-throated sloths (34/123) and 21.4% for Hoffmann’s two-toed sloths (41/192), while for mantled howler monkeys the survival rate of babies was slightly higher, 43.1% (41/95). This situation was due to many factors, such as weight, health status upon arrival at the JRC, and adapting to captivity without a mother figure.

When babies needed nursery care and could not be released at the rescue time, the rehabilitation process lasted for an average of 835.3 ± 109.5 days for Hoffmann’s two-toed sloths until they were entirely prepared for release as functional adults. However, 58.9% (113/192) of Hoffmann’s two-toed sloths died during quarantine, most of the cases being due to lesions caused by electrocution, trauma, etc., or because they were in their early stages of development, and 14.1% (27/192) died during nursery care before the release age. At the time of writing of the study, 5.7% (11/192) were still under nursery care at the JRC. In contrast, brown-throated sloths showed no survivability in captivity at the JRC: none of the offspring survived for longer than a few months, so their release was always attempted when they were considered viable animals at the rescue time. Therefore, only 34 releases were possible (28% of the rescued offspring). Finally, for the mantled howler monkeys, the average nursery time was 619.8 ± 192.1 days, with a decreasing tendency in the number of days that monkeys spent at the JRC observed thanks to improvements in animal husbandry techniques, such as the update of the protocols, diets, and ethological improvements. The successful release rate in mantled howler monkey offspring was 29.4% (28/95), while 13.7% (13/95) were still under nursery care at the JRC at the time of writing of the study. Mortality rates reached up to 41.1% (39/95) during quarantine, and 15.8% (15/95) died during nursery care before the release age.

### 3.3. Application of the Speaker Method

In the case of sloths, if the mothers reacted but moved slowly, the baby could be placed in a basket of natural fiber such as coconut, and the meeting could occur on the branch where she was. Then, if necessary, the rescue team members could climb up and help place the baby closer to the mother. However, this was an uncommon situation, and it was observed that generally the mothers climbed down the tree to pick up the baby (Appendix A). Hoffmann’s two-toed sloths are usually faster than brown-throated sloths, and the best time for the reunion was the evening. The best way to proceed was to place the baby on a branch; even newborns can hold onto an extension of the right size for their nails for a short period. The branch with the baby at the end was brought closer to the mother from a reasonably safe distance. This way, the mother had time to smell the baby before the JRC team and listen to its cry for proper recognition. Brown-throated sloths instead came down very slowly, sometimes taking hours. In these situations, the speaker was placed in the direction of the descending mother to direct her where she had to come down, as they would tend to choose incorrect paths that did not lead to the main tree trunk, or the tree was too wide for the animal to descend safely, and an adjacent tree proved more adequate. This technique usually involved waiting for the mother to come down to human height to show her the baby and see its reaction. If she recognized it, she would aim to take it in her arms and bring her face closer to smell it for recognition and acceptance (Figure 2). Mothers did not care about the surrounding person and focused on reclaiming their baby. 

Moreover, they climbed down quickly once they saw their baby. There were no vocalizations between them. The baby usually looked for the mother’s chest immediately, and the mother would not stop smelling the baby; there was typically lots of contact between their mouths for recognition, so minutes could pass until the mother decided to start the ascent. Usually, sloths needed a basket or someone to bring the babies closer to their mothers because they did not have enough agility. In the case of the brown-throated sloth, the basket proved essential as mothers did not come down to a close enough distance and could not get close to the baby. 

In mantled howler monkeys, the process usually developed fast. After observing and analyzing the potential dangers, the mother came to pick up the baby, as her caring instinct helped her overcome the risk. Therefore, for the reunion’s success, it was essential to leave the baby on a bed of leaves or a blanket of not too bright colors so as not to frighten the mother. Once reunited, the mother grabbed the baby in her arms, usually secured to the tree with her tail. She quickly climbed the tree again to a safe position, where she checked it and comforted the baby with hugs, contact between the two mouths, and comfort vocalizations. Moreover, as monkeys are more active and their development is faster than sloths, their reunion with the group was more straightforward. Their inherent mobility allowed them to climb trees when the group, attracted by the call, came.

### 3.4. Effectiveness of the Speaker Method

Due to survivability rates, only 21.4% of Hoffmann’s two-toed sloth (41/192) and 17.9% of brown-throated sloth (34/123) babies could be successfully released. All the released brown-throated sloths were reunited with their mothers as babies using the Speaker Method (100%, 34/34). In contrast, for Hoffmann’s two-toed sloths, only 31.7% (13/41) were released as babies, and 68.3% (28/41) were released after their rehabilitation stage at JRC (Figure 3 and Figure 4). Then, the Speaker Method was 91.9% and 45.8% effective for brown-throated sloths and Hoffmann’s two-toed sloths, respectively, as the technique was not adequate for 3 brown-throated sloths and 11 Hoffmann’s two-toed sloths that had to return to JRC due to either the mother’s absence or her unfortunate death. Using microchips allowed us to indirectly monitor each animal’s progress and potentially detect if they returned to the hospital for another admission. No incidents have been notified to the JRC concerning those animals released using the Speaker Method. 

For the mantled howler monkeys, the release could occur when they were babies, using the Speaker Method or not, or as adults. During the 5-year period, 29.5% of babies (28/95) were successfully released: 50% (14/28) using the Speaker Method, 46.4% (13/28) as adults, and for 3.6% (1/28) it was not required to use the technique as the baby was found with her mother. Both were immediately released together (Figure 5). 

## 4. Discussion

The role of WRCs in wildlife conservation is essential as an ex situ method with positive impacts at local and individual levels [31]. However, offspring survivability and effective release rates within WRCs are generally low, highlighting the need for novel methods to enhance them [32,33]. The present study describes the Speaker Method as an effective release technique for Hoffmann’s two-toed sloth, brown-throated sloth, and mantled howler monkey babies. 

Many reintroduction programs worldwide have successfully reintroduced offspring mammals to their wild group with different techniques [34,35,36,37]. In general, soft-release techniques are better for hand-reared orphans and species that suffer a significant impact from habitat loss [31,36,38]. The pre-release conditioning, acclimatization period to the release area, or supplemental feeding, among other techniques, increase the survival of the individuals [35]. Nevertheless, most offspring admitted to WRCs are rejected neonates or juvenile individuals. Still, rejection by the mother may happen due to stressful events (dog attack, car collision, etc.), previous experiences for the mother, or the social and environmental context [39].

Focusing on the species in the present study, the increase in neonate and juvenile sloths has obligated many WRCs to specialize in hand-rearing orphans to improve husbandry protocols [38]. To this end, it is essential to understand the effects that captivity can produce on wild-born individuals and the adaptative abilities of each species [31]. For example, if imprinted on humans, released animals may show no fear of them, and, being tame, may become susceptible to life-threatening experiences such as being hunted by humans [40] or losing their antipredator behavior [41]. Furthermore, orphaned mammals must prove to be able to feed, find water, mate, and recognize and defend themselves against their predators in the environment before being released [16,42]. Thus, the survival rates of hand-reared orphans depend on the species and their biological and behavioral requirements. Therefore, it is essential to carefully evaluate the feasibility of captive breeding programs for each species, considering their biology, behavior, and conservation needs [16]. The unique biology of sloths, in contrast to that of mantled howler monkeys, presents several challenges for their rehabilitation [30]. This can be observed in the high mortality rates of Hoffmann’s two-toed baby sloths (72%) and brown-throated baby sloths (78.6%) in captivity at the JRC. The differences between the two species could be due to the adaptability of Hoffmann’s two-toed sloth to captivity [38]. In contrast, the mortality rate of mantled howler monkeys is lower (54.7%), probably due to its biology and the implementation of new husbandry adaptations for this species, but it remains high.

Thus, considering these data, it becomes crucial for WRCs to implement a release method specifically designed for the early stages of mammal rehabilitation. In this context, the technique published by Miller (2007) to reunite juvenile and adult raptors inspired the development of the Speaker Method [20]. This new method allows the early return of healthy offspring to their families after a veterinary check, increasing their survivability and avoiding hand-rearing. This is a significant factor to consider since, following the IUCN Guidelines for Reintroductions, it is crucial to emphasize that this is the optimal timing for its reintroduction, thereby minimizing the potential mortality risk during rehabilitation under human care [43]. Our results confirm its efficacy in released baby mammals. When assessing “success” following the release of a wild animal, we should include monitoring post-release studies, including radio telemetry, GPS collars, or similar methods that allow us to evaluate the real success of each release [16,43]. Most of the studies conducted post-release monitoring confirmed the most significant percentage of failures observed during the initial four years, and these studies were conducted primarily on animal populations [44]. However, the lack of financial support for conservation studies makes it difficult to perform post-release monitoring due to elevated costs [31]. In the present study, all the individuals released using the Speaker Method carried a microchip to identify whether they were admitted to another WRC or found dead. As of the manuscript’s date, there have been no reported incidents in the past five years. It is worth highlighting that the microchips are specifically designed to identify and endure throughout the animal’s entire lifespan [45]. Using the Speaker Method can help save costs in the hand-rearing and rehabilitation of healthy babies until they grow up to a functional adult stage, so these resources can be used for monitoring studies, for example. Complete post-release monitoring studies are needed to assess the long-term survival of individuals released with the Speaker Method. 

Although the excellent acceptance between mothers and offspring in both species is noteworthy, it is crucial to recognize the essential differences between sloths and monkeys. Numerous investigations have been conducted on the similarities shared by these species and others, e.g., different species of New World monkeys, and their occurrence has been documented in multiple countries [4]. Therefore, applying this technique to several related mammalian species and diverse locations becomes a plausible consideration for future applications.

Finally, it is essential to highlight that even if WRCs focus on individual rehabilitation, they aim to serve the conservation of species in the environment. So, the success should also focus on the population’s health status [16]. The risk of introducing emerging pathogens to wild populations after animals having stayed at any WRC needs to be considered. Most emerging pathogens are zoonotic, so the One Health Perspective should be carefully considered in each situation [46]. The release of an animal into the environment should have no negative impact on the wild populations, such as the carrying of an emergent infectious agent or other harmful agents such as antimicrobial resistance genes or microplastics [16]. The early release of babies and their return to their mothers using the Speaker Method can minimize the transmission of pathogens to wild populations and increase the survivability of the rescued baby mammals.

## 5. Conclusions

In conclusion, despite challenges related to human interaction, the Speaker Method could be a viable early-release method for offspring of three different mammal species (Hoffmann’s two-toed sloth (*Choloepus hoffmanni*), brown-throated sloth (*Bradypus variegatus*), and mantled howler monkey (*Alouatta palliate*)). Although post-release monitoring studies are necessary to assess the survivability of these babies, the success of the reunion of the babies with their mothers suggests its potential for application with closely related species in different regions around the world. Furthermore, this method can be applied in rewilding other mammal species with similar characteristics. 

## Figures and Tables

**Figure 1 animals-13-03669-f001:**
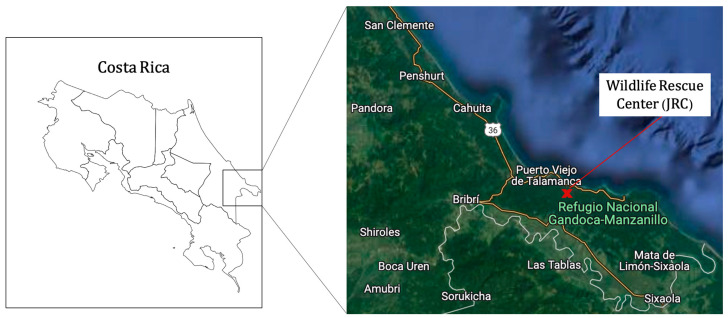
Map of Costa Rica showing the location of the rescue center and all the areas where the method was used.

**Figure 2 animals-13-03669-f002:**
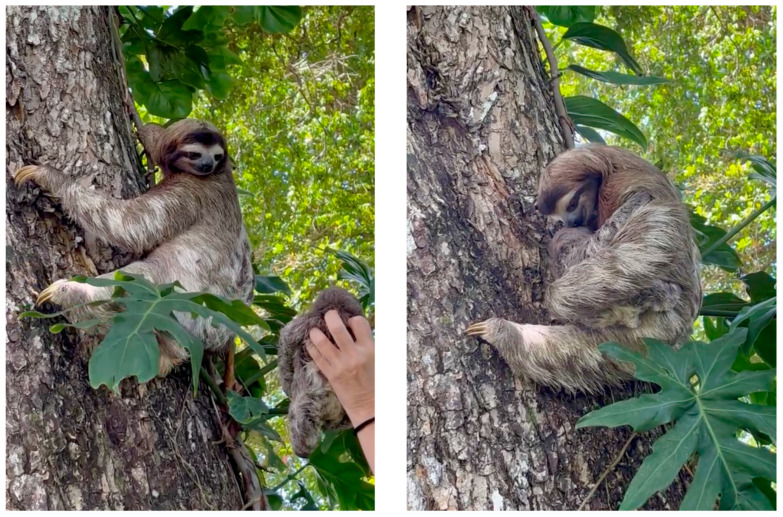
**Left**: Baby brown-throated sloth being presented to the mother who had climbed down the tree when she heard the cries reproduced on the speaker. **Right**: Female brown-throated sloth examining her offspring before engaging in the acceptance process. Notably, there are no discernible indicators of aggressive or hostile behavior.

**Figure 3 animals-13-03669-f003:**
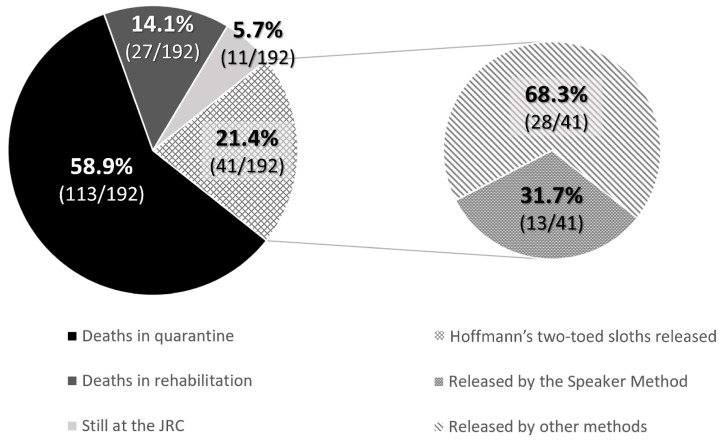
The proportion of released and deceased baby Hoffmann’s two-toed sloths between January 2018 and December 2022.

**Figure 4 animals-13-03669-f004:**
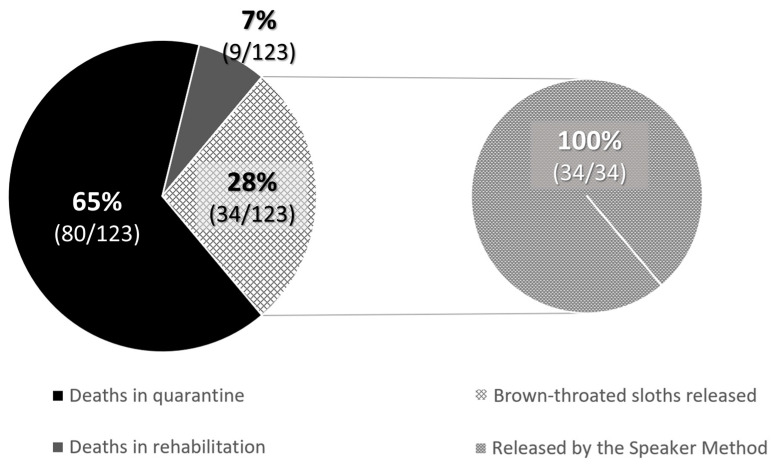
The proportion of released and deceased baby brown-throated sloths between January 2018 and December 2022.

**Figure 5 animals-13-03669-f005:**
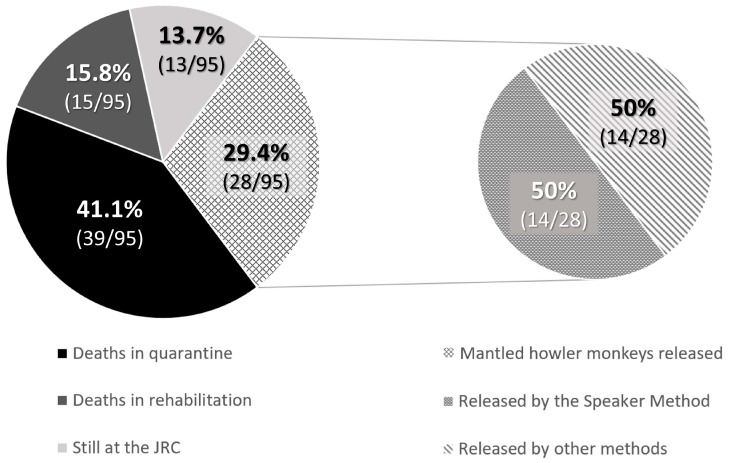
The proportion of released and deceased baby mantled howler monkeys between January 2018 and December 2022.

**Table 1 animals-13-03669-t001:** Captive survivability rates and data at the Jaguar Rescue Center between January 2018 and December 2022.

	Total			Babies		
	Released + Still in Rehabilitation	Rescued	% Survivability	Released + Still in Rehabilitation	Rescued	% Survivability
*Bradypus variegatus*	197	429	45.9%	34	123	28%
*Choloepus hoffmanni*	214	535	40%	41	192	21.4%
*Alouatta palliata*	94	254	37%	41	95	43.2%

## Data Availability

The data presented in this study are available on request from the corresponding author. The data are not publicly available due to the privacy of the rescue center.

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
