# Peer review of "The Speaker Method: A Novel Release Method for Offspring Mammals and 5-Year Study on Three Costa Rican Mammals"

_animals, 2023, doi:10.3390/ani13233669_

Round 1
Reviewer 1 Report (New Reviewer)
Comments and Suggestions for Authors
Dear authors,
I found your work very interesting, and very important since it deals with the work of WRC especially in a part of the world where high biodiversity takes place and many species that are treated in the WRC are threatened, vulnerable, or at the brink of extinction.
Let me be more specific.
In many WRC globally, many animals are released in the wild without any gps, and without following up the success of the release back in the nature.
As such, one cannot actually know if the release procedure was successful.
Whereas in the current paper, by using the "offspring call" towards the mother, and if the mother actually accepts the baby/subadult back, then the success of the release is guaranteed.
That fact, especially to the part of releasing threatened and vulnerable species back to nature, is a very important addition to the procedure.
The conclusions are consistent with the evidence and arguments presented and they address the question posed.
There is actually no statistics that takes place, but in that specific case there is no control-population easily, to conduct a statistical approach. Ι mean, there is no control in terms of having measured the rate of success of releases in the wild without the "calling" mehtod, to compare it now with the "calling" method.
Therefore, the simple descriptive statistics that take place here, are satisfactory.
I would only like to raise some points in need of improvement.
1.
In the summary you do not mention anywhere that all this happens within the context of the work of a WRC, and one stays a bit confused. Please include that in your summary.
2.
I miss some informative tables that would present the information better. At some points it is hard to follow up the text. I would suggest to include a table to present your results in "Captive Survivability"
3.
Please consider to cite an important work that outlines the importance of the work the WRC do in respect to conservation, you can find it here
https://doi.org/10.1016/j.jnc.2023.126372
4.
An additional improvement the authors could consider, are two
1 - gps tag all the offsprings that are released to the mothers, in order to be able to follow up in the long term how did the reunion between mother-offspring evolve once the release from the WRC with the "calling" method was over
2 - create very specific strict protocols, to which the authors make some references in the text, on how to proceed, approach, were to place the baby, how, how to play the playback. That should be written clearly, dilligently, in detail, as a procedure and be officially adopted by the organization.
If that would exist it could also be supplementary material to the paper.
5.
The figures are poor in quality, it would be great if they could improve the dpi's and enhacne the figures quality
Comments on the Quality of English LanguageI believe a moderate language/english editing is needed to your MS, to overall improve the quality of the text.
Author Response
Please see the attachment.

Reviewer 2 Report (New Reviewer)
Comments and Suggestions for Authors
This is a generally well-written manuscript (but see specific requests for clarification in places) presenting preliminary but timely results about the adjustment of a release technique in wild mammals that has been previously used in raptors. This technique involves the playback of the rescued offspring’s recorded calls to attract their mother in their original habitat. This research has key implications for restorative biological conservation with a focus on a critically endangered primate species.
The topic is well introduced and the background literature is appropriately reviewed and critically addressed, considering the paucity of the research in this specific domain. Overall, the layout is logical and the manuscript reads quite well. Findings from previous research are clearly summarized and put into perspective in light of the broader picture. I have no major issues with the outline/structure, conclusion, and overall content of the manuscript. The figures and underlying data are informative and relevant (but my request below about adding numbers to the figures).
This is an interesting study that clearly fits an information gap, and has significant implications for primate ecology, management, and conservation. On a side but important note, I want to praise the authors for their timely effort in seeking solutions to a major issue.
Before I can recommend this manuscript for publication, I just have a few questions/concerns, which I anticipate the authors will be able to handle easily.
My main content-based comment is about the potential for replicability of the methodological procedures. Details about the techniques employed should be more clearly and explicitly described.
CONTENT-BASED COMMENTS
Line 154-155 (and Line 181, Line 226-227, Line 298-299): “If the rescued animal is a juvenile, our protocol involves a thorough check to determine if the mother is still present within the rescue area.” – Please explain how this is done. My main question throughout your manuscript was: How can you be sure about the mother’s identity? Genetic testing? Tagging of both offspring and mother at the time of the rescue/capture? I think this question is particularly important in light of your comments in Line 197-201.
If there are cases in which you were not sure it was the mother being involved in the acceptance process, could your result inform research on adoption in wild mammals?
Line 175: “After the medical check-up, if there is a positive result,” – Please define what “a positive result” is in this case. This could help other researchers replicate your study.
Line 175-176: “the baby is compelled to cry to record the sound”. – Please explain how this is done (as this may have ethical implications).
Line 181: “placed up a tree into a basket as close to the mother as possible” – Same question as above: how do you know for sure this is the mother?
Line 184: “it was essential not to let her hear the baby crying simultaneously so as not to confuse her” – Please explain how this potential confusion was prevented.
Line 196: “they should be used only if strictly required” – Please explain how this was determined. Again, please think about the replicability of your study design and be as explicit as possible in your justifications to do or not to do certain things, and how exactly.
Line 209: “which has been observed only in sloths and monkeys.” – Could you please cite a couple of references for this statement?
Line 244 -245: “ranged from 194 grams (g) to 1300 g, with a mean of 50.88±32.89 g.” – This cannot be, can it? Could you please check and revise accordingly?
Line 255: “The successful releases without considering the age of the animals” – Please explain how success was determined, particularly in adult individuals that do not go through an “acceptance” process by their mother.
Line 275: “thanks to improvements in animal husbandry techniques” – Could you please be more explicit about these techniques?
Could you please add (maybe as an appendix) the detailed account of a few examples of unsuccessful attempts and your interpretation of what could have gone wrong? This would be particularly important in cases where obvious “mistakes” have been corrected and the subsequent attempts were successful.
Figure 3-5: Could you please add the ratios in brackets below each percentage?
Besides what you wrote about the timing of release in a diurnal versus nocturnal species (Line 177-179), did you expect the Speaker’s Method to work better with one of three mammal species under study, maybe due to specific socio-demographic and ecological characteristics, and within a certain offspring’s age range? If so, please explain the rationale for this prediction. Also, do you think your preliminary results based on three mammalian species can be generalized to other mammalian species and if so, on what basis? These suggested additions could be placed in the Discussion section.
Comments on the Quality of English LanguageFormat-based comments
Line 90: “This is why is essential to develop” – Grammatical approximation.
Line 97-101: “Although it is possible to breed and reintroduce orphaned animals into the wild successfully, it is preferable, whenever possible, to return them to their mother, as their offspring stage is essential for correct learning and behavioral development. Individuals deprived of these interactive contacts may show motor, biological, and behavioral deficiencies [16].” – These sentences are almost identical to those in Line 85-89. Please revise to avoid such obvious redundancies.
Line 121: Define the “JCR” acronym when you use it for the first time. Also, isn’t there a typo here? Is JCR the same thing as “JRC” (Line 125-126)?
Line 127: Remove full stop after closing bracket.
Line 165-167: “The survival of this species if kept in a WRC is negligible being the procedure of returning the baby to the mother a key step in the rehabilitation of these animals.” – Sorry, I don’t understand this sentence. It’s probably due to its grammatical construction. Could you please clarify?
Line 179: “hhe speaker” – Typo.
Line 191-193: “The use of blankets, baskets, or objects that could appease the mother's attack 191 was dangerous; neither the person who climbs the tree could be used because she would 192 feel threatened and would attack.” – Sorry, I don’t understand these two sentences. It’s probably due to their grammatical construction. Could you please clarify?
Line 227: I think “accept” should read “accepted” (to be consistent with the past tense of the other verbs in this sentence).
Line 230: “tu the JCR” – Typo.
Line 267: “lesions led from electrocutions” – Do you mean “caused by”
Line 267: “very short age” – Grammatical approximation.
Line 273: “monkey” – plural? (same comment Line 276 and Line 344).
Line 296-297: “an adjacent tree resulted more adequate” – Grammatical approximation.
Line 344: “the release could be as babies” – Grammatical approximation.
Author Response
Please see the attachment.

This manuscript is a resubmission of an earlier submission. The following is a list of the peer review reports and author responses from that submission.
Round 1
Reviewer 1 Report
Comments and Suggestions for Authors
I did not understand why The Speaker’s Method was called this way until the methods. A brief description when it is mentioned in abstract and intro would help the reader.
References missing for Costa Rica wildlife conservation status, please add.
Line 93 - you mean flora and fauna
Lines 172-185 - this does not seem very safe for the offspring
I do not understand why the method is different from reuniting the mother with baby at the beginning of the rescue. You say at line 143 that mothers often reject offspring so why would they not reject while the method is used?
How is this method validated? It is not clear from the methods. I would suggest you remove the word validation (which implies following up on survival and data analysis) and just keep a descriptive approach for the method. You can still talk about percentages.
Line 212 - you mean descriptive statistics
Comments on the Quality of English LanguageThe quality of English language is average and not always suited to scientific writing. Some sentences are clunky and need removing (e.g. lines 22-23), shortening (e.g. lines 19-21) or rewriting (e.g. lines 37-38).
The method is not actually been validated.
Reviewer 2 Report
Comments and Suggestions for Authors
I seriously think the authors should consider their title as to me it makes no sense what so ever. Prepare something that is easily understandable for a layperson and that makes it immediately clear what the study is about.
Simple summary – what is the Speaker’s Method? This seems to be key, but I am none the wiser having read the summary. What is the efficacy rate you refer to? Also give a bit more detail on what it is exactly that you found.
Abstract – more or less a repetition of what I describe above, what is the Speaker’s Method? What is the efficacy rate you refer to? Here it is even more important that you give more detail on what it is exactly that you found.
L37 I think this is too general an Introduction; also you single out freshwater species but it is clear from your paper that that is not your focus. Present some more relevant statistics here. I think this should be replaced by a much more detailed and balanced overview of the role that rescue centres play, and how successful they are in reintroducing specific species.
L42 critically endangered – change to Critically Endangered; Also Endangered and Vulnerable.
L53 – now suddenly you present data on birds – why?
L72 – this seems to be a very important study and part of your rationale of conducting the study – explain this in more detail.
L77 – expand the aims and make them all very explicit
L83 What is JRC? Where is it based. [I see this is explained later in L90 – but reorganise this]
L117 – here it is listed a Least Concern, whereas in L44 it is stated that it is listed as VU.
L151 specie change to species.
L175 genders – change to sex.
L227 – you never define what you consider a successful release. Did you actually monitor the released animals? If so, for how long? And how did you do it (as it must be difficult to follow a released animal without collaring it or tagging it in another way).
L229 – how do you define survivability – in captivity it is easy but how does that translate to the released animals.
I actually think that you cannot make any statement on whether a release was successful as you simply did not follow up on it (as indeed indicated in the Discussion).
Discussion
See what I wrote for the Introduction – I expect to see a much more detailed and balanced discussion on the survival rates of released animals (there are some excellent studies out there where they actually did monitor post release). Now the focus is solely on your own findings and I would expect a broader perspecitvie.
L385 – the conclusions are largely repetitive and do not provide any support for the statement made; as such it does not add any value to the paper
L416 – the references need work and are not in the format needed for Animals.
Comments on the Quality of English LanguageIt is largely alright but there are some sections that need to be looked at.